# Methods for the Assessment of NET Formation: From Neutrophil Biology to Translational Research

**DOI:** 10.3390/ijms232415823

**Published:** 2022-12-13

**Authors:** Marina Stoimenou, Georgios Tzoros, Panagiotis Skendros, Akrivi Chrysanthopoulou

**Affiliations:** Laboratory of Molecular Hematology, Department of Medicine, Democritus University of Thrace, 68100 Alexandroupolis, Greece

**Keywords:** neutrophil, neutrophil extracellular traps, techniques, tools

## Abstract

Several studies have indicated that a neutrophil extracellular trap (NET) formation, apart from its role in host defense, can contribute to or drive pathogenesis in a wide range of inflammatory and thrombotic disorders. Therefore, NETs may serve as a therapeutic target or/and a diagnostic tool. Here, we compare the most commonly used techniques for the assessment of NET formation. Furthermore, we review recent data from the literature on the application of basic laboratory tools for detecting NET release and discuss the challenges and the advantages of these strategies in NET evaluation. Taken together, we provide some important insights into the qualitative and quantitative molecular analysis of NETs in translational medicine today.

## 1. Introduction

Neutrophils represent the most abundant innate immune cells and are the first to migrate to sites of tissue inflammation. Their half-life in circulation varies from 6 h to a few days. Thus, at steady-state conditions, renewal and maintenance of the neutrophil pool are ensured by constant bone marrow granulopoiesis. However, during systemic inflammatory conditions, a reprogramming of the hematopoietic response, named emergency granulopoiesis, leads to the commencement of de novo generation of high numbers of neutrophils from myeloid progenitors and their mobilization to circulation [1].

Neutrophils contain a multilobular nucleus and have a 12–15 μm diameter [2]. The different types of granules include (a) the primary/azurophilic, (b) secondary/specific and (c) tertiary, which differentiate them into three distinct categories. The primary granules contain myeloperoxidase (MPO), defensins, elastase and cathepsins, while the secondary granules contain NADPH oxidase, lactoferrin and gelatinase. Finally, the tertiary vesicles consist of alkaline phosphatase and gelatinase [3,4].

The traditional concept that neutrophils are terminally differentiated cells with limited plasticity and highly conserved function due to their low transcriptional activity has been critically revised. Recent studies have provided evidence suggesting that neutrophils express a wide variety of surface receptors that give them the ability to quickly respond against disease–environmental cues and undergo transcriptional reprogramming, allowing them to acquire disease-specific phenotypes [5,6]. Thus, neutrophils are considered a cell population with diverse functions, plasticity and a longer survival time of a few days than the one initially suggested of 8–10 h [7].

High-throughput approaches have revealed that neutrophil gene expression includes inflammatory agents, such as cytokines and chemokines, that complement the components [8,9] and factors involved in inflammation resolution [10]. Neutrophils interact with a variety of cell populations during their migration toward tissues, including endothelial cells, fibroblasts, macrophages, lymphocytes and dendritic cells [2].

In the early 2000s, advances in molecular biology proposed the most important update of neutrophil physiology, particularly their capacity to release neutrophil extracellular traps (NETs) to kill bacteria [11]. Activated neutrophils generate NETs, which are extracellular fibers composed of chromatin, histones and granular proteins [11]. A NET release is a mechanism through which a neutrophil releases NETs and eventually dies [12]; however, a NET release is not always linked to the death of the neutrophil [12]. Overall, there are three different types of NETs (Figure 1). In the first type, NETs are composed of mitochondrial DNA together with granular proteins, and neutrophilic death does not follow (Figure 1) [13]. The other two types of NETs are called vital and suicidal. The suicidal mechanism of NETs is dependent on the NADPH oxidase-2 enzyme and is accompanied by the death of neutrophils (Figure 1) [12]. The activation of NADPH oxidase relies on an increase in the Ca^2+^ concentration in the cytoplasm and sometimes on the generation of ROS in mitochondria [14]. On the other hand, vital NETs are independent of NADPH oxidase-2, and neutrophils remain alive (Figure 1) [14]. A NET release can be induced in vitro by either inflammatory molecules (i.e., cytokines, pathogen components) that simulate the disease microenvironment or chemical compounds (e.g., PMA, ionomycin) that activate some intracellular pathways [15,16,17]. Each stimulus may induce a different pathway of NETs (Figure 1) [12]. The morphology of NETs, depending on the available space, is either wide or cloud-like [15,18,19].

Once neutrophils are activated, several downstream cellular events occur. During NET formation, neutrophil elastase (NE), peptidyl-arginine deaminase 4 (PAD4) and myeloperoxidase (MPO) migrate to the nucleus. Indeed, PAD4 binds with calcium, and the complex is then transferred to the nucleus, where histone hypercitrullination and DNA decondensation are mediated [14]. The cytoskeletal components and nucleus envelope, as well as chromatin, are disintegrated by serine proteases. Finally, DNA, citrullinated histones (CitH3) and granular proteins are released into the extracellular environment through pores on the cellular membrane (Figure 2). 

Since NET formation was first reported in 2004 as an eliminating mechanism for invading microbes [11], NETs have attracted much interest, and several of their additional functions have been discovered. Another important role of NETs is the activation of other immune cells during inflammation, either to orchestrate an inflammatory response or to resolve inflammation [20]. Aggregated NETs contribute to degrading cytokines and chemokines, resulting in inflammation resolution in the acute inflammatory response of gout [21].

Accumulating evidence suggests that NETs induced by different stimuli and disease conditions may be heterogeneous in regard to protein composition and post-translational modifications acquiring different biological effects [16,22]. In addition to DNA and granular proteins, NETs contain disease-related proteins that have a specific role in numerous human disorders. In particular, NETs have been proposed to induce the formation of autoantibodies in diseases, such as small vessel vasculitis and systemic lupus erythematosus (SLE) [23], as well as acting as functional scaffolds for thrombosis by aggregating platelets [24]. Furthermore, NETs participate in chronic inflammation [25] and fibrotic disorders via cross-talk between NETs and mesenchymal cells [26]. 

Because of this vital clinical significance, the role of neutrophils/NETs as a pivotal therapeutic target in inflammatory pathology has been reconsidered. Therefore, either the spontaneous formation of NETs (ex vivo biological sample) or the in vitro-induced NETs should be easily and efficiently studied by the application of the appropriate laboratory methods. Here, we present and discuss the updated tools available for quantitative and qualitative evaluation of this mechanism.

## 2. Tools for the Assessment of NETs

### 2.1. Enzyme-Linked Immunosorbent Assay (ELISA)

ELISA is a commonly used tool in NET research, allowing NET detection and quantification. This method provides clinical significance by associating NET markers with human disorders, such as cancer [27], autoimmune small-vessel vasculitis [28,29], rheumatoid arthritis [30], ulcerative colitis [31], lupus [32,33,34], SARS-CoV-2 infection [35,36] pulmonary [37,38] and cystic fibrosis [39,40]. There are several validated, commercially available ELISA tests for detecting NETs [41], as well as in-house sandwich ELISA assays [42].

The scientific community is still in controversy regarding which is the most targeted biomarker for detecting NET formation with the ELISA technique. In particular, detecting nucleosomes and circulating cell-free DNA does not always indicate NETs since they may be derived from other events, such as apoptosis or necrosis [43]. Similarly, several ELISA kits have neutrophil-derived enzymes, such as myeloperoxidase and neutrophil elastase, as NET markers that may not accurately reflect neutrophil activation, degranulation and NET formation [44]. 

In 2017, an ELISA was presented for detecting citrullinated histone H3 (CitH3) in human plasma. While the assay detects NET formation with high specificity, CitH3 as a NET biomarker is specific only to the PAD4-dependent pathway [45]. Two years later, an optimized nucleosomal CitH3-DNA ELISA was validated for reliable NET quantification in human plasma [46]. Complexes of extracellular DNA with myeloperoxidase (MPO) [29] and neutrophil elastase (NE) [44] are considered specific NET biomarkers. Methods that detect the presence of two markers are better. However, a recent study alerted the NET community about the specificity of MPO-DNA complexes with ELISA [47]. An MPO-DNA ELISA proved to be specific for NET formation in vitro but was unspecific to in vivo and no correlation between plasma MPO-DNA complexes and other NET biomarkers were found, suggesting the need to use isotype control antibodies and additional NET markers [47]. 

The significant advantage of NET quantification by ELISA is the availability of several ELISA kits, as it is easy to use [41]. Furthermore, NET detection by ELISA is an uncomplicated procedure with high specificity and accessible equipment by several laboratories [48]. Compared to other techniques applied in NET research, ELISA assays have a low cost (Table 1) [41]. Nevertheless, NET quantification by ELISA requires assay standardization, as several employable ELISA kits offer limited reproducibility of the results [46]. An additional challenge is the evidence of a restricted correlation between detecting NET formation by ELISA in vitro and in vivo (Table 1) [46,47].

### 2.2. Immunofluorescence Microscopy 

NET visualization by immunofluorescence microscopy is a commonly applied methodology based on immunostaining of NET components, such as granular staining for neutrophil-associated enzymes and nuclear staining for histones and extracellular DNA, allowing the visualization of their expression (Figure 2 and Figure 3) [11]. The basic strategy of this methodology to detect NETs is using DNA intercalating dyes to visualize DNA and specific immunostaining for the enzymatic components of NETs to improve quantification and visualization (Figure 2 and Figure 3) [15]. Using extracellular DNA and neutrophil-derived proteins (e.g., CitH3, MPO, NE etc.), a NET formation can be detected by immunofluorescence microscopy in several biological specimens, such as tissue sections, peripheral blood samples, sera, bronchoalveolar lavage fluids and cell culture mediums (Figure 2 and Figure 3) [53]. Immunofluorescence microscopy can characterize NETs in vitro as a response to pathogens or chemical agents in tissue culture plates, ex vivo in blood samples and in situ in histological samples from humans or animals [15].

A great benefit of detecting NETs by immunofluorescence microscopy is the visible differentiation of the distinct stages of NETs based on the morphological changes of the nucleus [50]. Furthermore, immunofluorescence microscopy allows visible differentiation between NETs apoptosis and necrosis [50]. Immunofluorescence staining of virulence factors enables the visualization of them entrapped in NETs [11]. Another strength of this technique is that specimens treated with immunostaining can be stored for up to six months at 4 °C (Table 1) [50]. 

However, detecting NET release by immunofluorescence microscopy can be observer-dependent based on the field of view [15]. Apart from being laborious and time-consuming, the quantification of NET release is prone to errors with limited data reproducibility due to subjectivity [52]. Another drawback is the intensive precaution during sample fixing and staining [62]. Furthermore, this technique is suitable only for analyzing a small number of cells in a sample (Table 1) [15].

As mentioned in the Introduction section, NETs are also enriched with a variety of disease-related proteins closely related to the pathophysiology of each disorder (“disease-related” proteins) [16]. For instance, bioactive tissue factor (TF) was detected on NETs in vein and arterial thrombosis [24,63] and mature interleukin-1 beta (IL-1β) in Familial Mediterranean Fever [64] and gouty arthritis [65]. Hence, in a disease model, the expression of “disease-related” proteins on the NET scaffold can be studied by applying specific antibodies that target them. In order to concomitantly verify that these proteins are derived from neutrophils that are undergoing NET formation, specimens could be co-stained with robust markers of NETs, including at least two antibodies against neutrophil-derived proteins (CitH3, NE or MPO) and DNA intercalating dyes (Figure 3) [51,66,67]. 

A selected example for the assessment of NET formation by immunofluorescence microscopy is the study of Brinkmann et al., who validated a semi-automatic protocol for NET quantification in vitro. Specifically, it involved using antibodies against a subnucleosomal complex (histone 2A, histone 2b and chromatin) and neutrophil elastase. The number of cells per field was determined by staining cells with the DNA-intercalating dye, Hoechst 33342. This method is based on the observation that anti-chromatin antibodies can more easily and directly bind to decondensed chromatin, characterizing the cells that undergo NETs. A correlation was identified between the fluorescence signals of the anti-chromatin antibody and the signals of DNA-binding dye, resulting in the calculation of the percentage of netting neutrophils [51]. Data interpretation of this technique does not require complex equipment, only a fluorescence microscope and a public-domain software package [51]. Furthermore, Coelho et al. used an antibody against the complex of histone-DNA and DAPI for DNA staining, detecting NETs with immunofluorescence microscopy and open-source software [52]. Similarly, von Köckritz-Blickwede et al. used an antibody against the complex of histones-DNA and DAPI, detecting NETs with immunofluorescence microscopy; however, they examined the samples only with the aid of a microscope without using software [50]. Notably, the quantification of NETs with software-based methods is unbiased, objective and valuable for data standardization and comparison [51].

At the time of this writing, the 3D9 monoclonal antibody against the cleavage site of histone H3 (H3R49) has been reported as a novel tool for detecting and quantifying NET formations in humans by using immunofluorescence microscopy in purified neutrophils and tissue sections [68]. Although ex vivo validation of this method in serum/plasma is pending, it appears to be a specific marker of NETs and distinguishes NETs from the chromatin of other cell fractions and neutrophils that die via alternative mechanisms, such as apoptosis, necrosis and necroptosis [68].

### 2.3. Electron Microscopy

Another method for NET visualization is electron microscopy. It is a quite common and relatively old technique based on the acceleration of electrons to visualize a sample placed in a vacuum chamber [69]. Both transmission electron microscopy (TEM) and scanning electron microscopy (SEM) methods are based on an electron beam, either passing through or reflecting from the surface of the sample. In TEM, the electron beam passes through the sample and is projected onto a fluorescent screen, producing an image [70]. In contrast, in SEM, the electron beam reflects upon one area of the sample and the emitted signal is then collected by detectors where an image is produced [54,70]. Additionally, TEM is mainly used for visualizing the inner parts of a sample, whereas SEM is used for the surface examination of a sample [54]. TEM provides a higher image resolution with the image projected in 2D, while SEM yields a greater depth of field, thus producing a 3D image [54,70].

Electron microscopy is a beneficial methodology to assess NETs (Table 1). According to the literature, SEM seems to be the central choice for visualizing NET formation. In general, the electron microscope provides a better image resolution and magnification compared to conventional light microscope visualization [54]. Thus, NETs can be differentiated from fibrins when high-resolution SEM is applied [20]. Nevertheless, the high energy projected from the electron beam can sometimes change the structure of a sample and therefore provide a faulty image. That is why special preparation of the sample is required, which can often be a very complex and expensive procedure (Table 1) [54].

Conventional electron microscopy staining methods still cannot visualize NET formation as accurately. There is a reliable protocol where TEM can also be used to demonstrate NETs and interactions with bacteria [71]. Nevertheless, TEM is a distinctive visualization method for autophagy [71], as it allows the visualization of the inner parts of a cell. Autophagy, which is closely associated with the release of NETs [18,72], is an intracellular mechanism and can therefore be visualized in great depth. The work of Tang et al. demonstrated the increased vacuolization that neutrophils undergo when treated with H4B4 (an anti-human LAMP-2 antibody), which in turn led to an increase in NETs [73]. Electron microscopy was also used in the work of Fuchs et al., where an interesting connection between NETs and platelets was demonstrated where NETs played a promoting role toward thrombosis [24]. In addition, Ermert et al. used SEM to observe NET formation in neutrophils isolated from mice, while Manzenreiter et al. investigated the complex role of NETs in cystic fibrosis by SEM [74,75]. In more recent work, Rajeeve et al. investigated NET formation through the infection of neutrophils with Chlamydia trachomatis, which normally blocks NET production, but when the chlamydia protease CPAF is absent from the bacteria, NETs are observed [55].

### 2.4. Live Imaging

Intravital microscopy allows imaging of immune responses at a molecular level while the animal is alive, usually involving either confocal or multiphoton microscopy [76]. The basis of multiphoton microscopy is photon-induced fluorescence excitation and subsequent electron emission detection. It has several advantages compared with confocal microscopy, as it allows deeper tissue penetration and involves less phototoxicity [76]. In contrast to confocal microscopy, it does not require a pinhole to reject out-of-focus light but has inherited optical capability [76,77]. However, this is solved with the addition of the spinning disk to the confocal microscope, which allows the generation of multiple excitation points at the same time and not just one, leading to reduced phototoxicity [56,78].

Specifically, regarding NETs, it allows real-time NET visualization. The basic principle of this methodology is staining components that indicate NET release to identify their location and visualize their structure [79]. Furthermore, intravital imaging enables the close examination of the NET structure (Table 1). However, the application of this technique to identify NETs in tissue specimens may demand the staining of several distinct components of NETs due to overlapping antibodies and signal emissions from different cellular components (Table 1) [79].

Kang et al. applied this technique to visualize NET formation with the detection of extracellular DNA using Sytox Green in the peri-infarct cortex of mice and associated NETs as toxic signals for the remodeling process that occurs after a stroke [80]. Another study involved the use of spinning disk confocal intravital microscopy to visualize NETs by staining neutrophil elastase (NE) and Ly6G in obese mice, where NETs are impaired due to obesity because of the presence of dysfunctional platelets [81], which have been shown to play an important role in sepsis [82]. This study aimed to gain further insights regarding the obesity paradox in sepsis, showing reduced inflammation in the liver of obese mice [81].

Hoppenbrouwers et al. also used this technique in an effort to assess the robustness of different NET inducers when stimulating neutrophils from healthy individuals with different molecules in vitro [57], including the organic molecule phorbol 12-myristate 13-acetate (PMA), ionomycin, lipopolysaccharide (LPS) and gram-positive or -negative bacteria [57]. Their study has suggested PMA, ionomycin and bacteria as strong inducers of NETs. Their use of time-lapse imaging also revealed a different series of events when comparing the NET process triggered by ionomycin and PMA [57].

A novel method for a quick, high-throughput and reproducible quantification of NET release in real-time is the use of the IncuCyte ZOOM Imaging Platform [83]. This method can identify NETs from other cell death pathways by detecting morphological changes in cells, nuclei and kinetics [83]. Since the IncuCyte ZOOM Imaging Platform permits the use of only two different fluorescent signals, it limits the ability to detect additional neutrophil components. Overall, this platform uses three imaging channels: the red fluorescent channel, the green fluorescent channel and the phase contrast channel [83]. The requirement for complex equipment and the potential for false-positive results when the plasma membrane integrity is compromised are additional drawbacks of this technique [83].

### 2.5. Flow Cytometry

Flow cytometry is an attractive tool for evaluating NETs that characterizes single-cell morphology, size and granularity [58]. The basic principle of flow cytometry relies on light scattering and fluorescence emission from dyes or conjugated antibodies, which bind to cytokines, receptors, cellular components and DNA, leading to the differentiation of distinct cell populations in whole blood (Figure 4) [58].

Some well-established protocols in NET research include staining with fluorescent dyes or monoclonal antibodies specific to cell surface markers or intracellular markers of NETs. Key components of NETs could be detected using flow cytometry with fluorescently labeled antibodies together with DNA dyes indicating NETs (Figure 4) [49].

Gavillet et al. developed a flow cytometry protocol, detecting MPO and citrullinated histones in vivo, directly in whole blood samples, without the need to isolate neutrophils. Thus, this flow method allows rapid assessment of NETs and quantification in a large cell population [49]. This methodology applied the detection of netting neutrophils both in vivo in clinical blood samples and in in vitro conditions [49]. In 2017, Masuda et al. proposed a quantitative, simple Flow Cytometry-assay that detected NETs by using a DNA-binding dye (Sytox Green), but it was unable to pass through the membrane, indicating that green-stained cells underwent NETs [59]. Zharkova et al. suggested the application of a fluorescently labeled antibody mixture against neutrophil cell surface markers [84]. This allows the direct detection of NETs in a mixed population.

Flow cytometry allows rapid screening of a large population of neutrophils in a small sample volume and provides detailed qualitative and quantitative information with high efficiency in a short time [41]. Additionally, this methodology allows NET quantification with high objectivity [85]. Hence, this technique is a notable diagnostic tool, associating NET generation with several human disorders (Table 1) [85]. However, the reproducibility of the results requires extensive precaution during sample preparation [84], which is the main drawback of this technique. Given that flow cytometry detects intact cells, this technique allows the detection of NETotic potential during the early stages, i.e., pro-NETotic cells, before they die by forming NETs (Table 1).

### 2.6. Multispectral Image Flow Cytometry (MIFC)

Multispectral Imaging Flow Cytometry (MIFC), based on immunofluorescence and fluorescent-activated cell sorting (FACS), can semi-automatically quantify NET formation in several samples [60]. In 2015, Zhao et al. made use of MIFC to measure both “suicidal” and “vital” NETs, investigating the morphological changes of netting neutrophils [86]. Imaging of neutrophils in a fluid stream provides information about the nuclear changes, allowing the characterization of the cell death mechanism since the formation of blebs can be clearly distinguished [86]. In the study of Zhao et al., MPO was co-localized with DNA in “suicidal” NETs.

Apart from distinguishing the cell death process (“suicidal” NETs, “vital” NETs and apoptosis), MIFC adds several additional benefits to NET research. This technique provides simplicity and reproducibility of quantitative results while being available for several sample analyses [60]. Detecting NETs by MIFC automatically and rapidly might enable the use of NET formation as a biomarker of human diseases and chemical compound screening to investigate the reflection on the NET formation (Table 1) [86]. However, the primary disadvantage of this technique is the capture of only netting neutrophils in the early phase, which might underestimate neutrophils in later phases of this mechanism [86]. Furthermore, MIFC may miss neutrophils undergoing “vital” NETs [86], and this technique requires sophisticated equipment for data acquisition (Table 1) [86].

### 2.7. Western Blot

Western blot or immunoblotting is a technique that can be used for NET detection. In the same context, disease-specific proteins exposed on NETs could also be detected by immunoblotting.

Similar to ELISA, it is a frequently used technique for antigen detection [87]. Specifically, with Western blot, a protein or antigen can be detected among multiple proteins/antigens in a host. The process, in brief, involves the separation of proteins based on the size in an SDS-polyacrylamide gel and the subsequent transfer of these proteins to a nitrocellulose membrane. A primary antibody specific to the protein of interest is added to this membrane. Detection is performed either directly, meaning that the label will be on the primary antibody, or indirectly, meaning that a secondary labeled antibody will be added [87].

The markers used to detect NETs are mainly extracellular and modified histones, such as H2A/H2B or CitH3. PAD4 can also serve as a marker. Pulavendran et al. used labeled antibodies against these specific markers to evaluate NETs in BAL fluids isolated from mice infected with a strand of bacteria called *Francisella tularensis*, which is the cause of pneumonic tularemia. The infected mice showed higher levels of NETs than healthy mice [61]. Liu et al. also used CitH3 as a marker to assess NET formation with Western blot in additional experiments of a phenomenon called ALI, or acute lung injury, mediated by LPS. In particular, they isolated lung tissue from mice treated with LPS and noticed an increase in CitH3. Therefore, NETs were present in the tissue of these mice rather than in the control mice. These NETs could represent the source of organ damage and the initial inflammatory reaction. Notably, the activation is not direct, as LPS alone cannot efficiently induce NET formation, but platelets that are activated by LPS can indeed make this happen [88]. Similarly, our group has also applied CitH3 as a protein marker during the immunoblotting process to support that fibrosis-related agents are able to trigger NET release in neutrophils obtained from healthy individuals. Indeed, our data further suggested that NETs generated upon this treatment are involved in the ensuing fibrosis, providing a link between NET formation and human fibrotic diseases [26].

Western blot is a particularly beneficial technique because it is sensitive and specific and provides some quantitative results based on the intensity of the bands (Table 1). However, there are some disadvantages, such as the significant limitation of the need for a primary antibody; otherwise, the technique cannot be performed. The most common problem is the antibody’s required concentration that is needed to detect the specific protein efficiently and not have background staining. These antibodies are also not very cheap and often lose their specificity when thrown into a mix of many proteins. Finally, the protocol is not standard, and the optimization process can sometimes prove difficult since Western blot is a very tender and multi-step technique (Table 1) [89].

### 2.8. DNA Dyes

A rapid and easy method to detect NETs in a plate assay employs impermeable DNA dyes, such as Sytox Green. This assay’s principle relies on the fact that the cell membrane is intact before NET formation. An impermeable DNA dye (e.g., Sytox Green) can be used to quantify the released cell-free DNA, including the formation of NETs as they are present in the extracellular space. A permeable DNA dye (e.g., pico-Orange) can also be used to quantify the total DNA, including intact cells, in order to measure a ratio of NET/total DNA. If they have different emission spectrums and different region binding, then they can be used at the same time in the same well. That means that one would have to bind in the major grooves and the other in minor grooves. Quantification of the binding requires a fluorometer [60]. Yost et al. utilized Sytox Orange, an impermeable dye, to assess NETs, where neutrophils isolated from adults and neonates were stimulated with LPS, and it was shown that (particularly in adult samples) extracellular DNA was higher; hence, a higher number of NETs was present. Of course, they still used microscopy techniques with both an impermeable (Sytox Orange) and a permeable dye (Syto Green, not to be confused with the impermeable Sytox Green DNA dye) to further verify the induction of NETs a marker for NE was also used [90]. Another more recent work by Hopke et al. regarding neutrophil swarming and NET release against C. albicans, Sytox Green was used to visualize the swarming effect and NETs. By measuring the fluorescence intensity of Sytox Green and the number of nuclei, they confirmed that NETs were released after swarming the fungi, which is an event that corresponds to the almost simultaneous recruitment of neutrophils against a pathogen [91].

The main advantage of this technique is that it is quick and easy, where quantification of DNA release and, therefore, NETs are available. On the other hand, it should not be used as the only method to quantify NETs and is also not very reliable since any effect/stimuli that disturb the integrity of the cell membrane will give false-positive results. Therefore, distinguishing between NETs and necrosis or apoptosis is not possible unless more markers are used [60]. In other words, cell-free DNA could originate from cells other than neutrophils, and hence, there is a need to engage further in an ELISA assay able to concomitantly measure extracellular DNA and a neutrophilic marker, such as myeloperoxidase (MPO) or neutrophil elastase (NE) [26].

## 3. Concluding Remarks

The increasing implications of NET formation in neutrophil biology and translational medical research require reliable and efficient tools for ΝΕΤ identification. Immunoassays, such as ELISA and Western blot, are used for NET quantification, but the results require standardization (Table 1). ELISA is a widely available technique due to its low cost, increased accessibility and simplicity. Western blot can have high costs and be laborious because of the extensive sample preparation. Flow cytometry-based approaches allow NET quantification with high objectivity and reproducibility in a rapid manner. However, sample preparation can be challenging. In addition, flow cytometry identifies and analyzes pro-NETotic cells prone to release NETs (Table 1). Visualization methods effectively distinguish the cell death mechanism. Nevertheless, immunofluorescence and electron microscopy are highly dependent on the observer and the field of view (Table 1). Therefore, it seems that there is no gold-standard method for the assessment of NET formation. Until today, a combination of different molecular techniques should be used to avoid possible misinterpretation of the results and provide the utmost comparable data. Moreover, functional assays are used to demonstrate a definite biological role for NETs and various NET-located proteins that are detected in the circulation and/or tissues. Considerably, the clinical significance of accurate NET detection and the application of molecular biology techniques in this field needs to be standardized and optimized further.

## Figures and Tables

**Figure 1 ijms-23-15823-f001:**
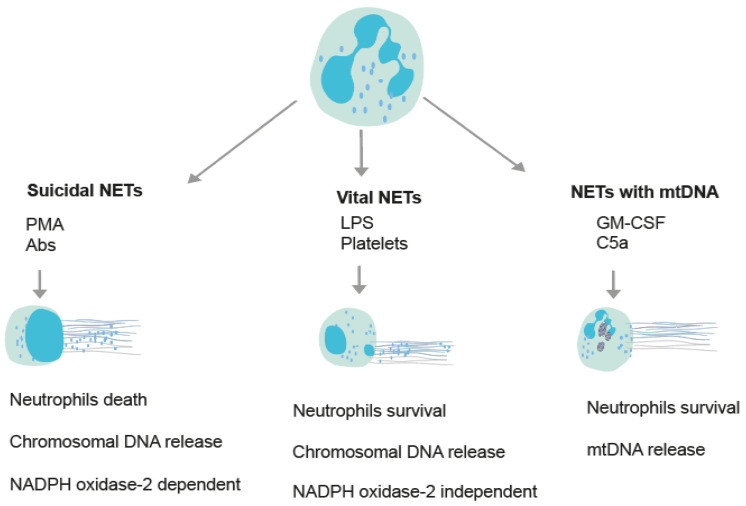
Schematic diagram demonstrating the different types of NETs. Depending on the stimulation, various molecular pathways result in the generation of NETs. This diagram presents the fundamental pathways of NET production, and in particular, the vital NETs or NADPH oxidase-2 (NOX) -independent pathway and the suicidal NETs or NADPH oxidase-2 (NOX) -dependent pathway. Furthermore, NETs can also consist of mitochondrial DNA (mtDNA) rather than chromosomal DNA. In this pathway, intracellular grey structures represent mitochondrial organelles. Some indicative NET inducers are chemical compounds or inflammatory agents (Phorbol-12-myristate-13-acetate (PMA), antibodies (Abs), lipopolysaccharides (LPS), activated platelets, granulocyte-macrophage colony-stimulating factor (GM-CSF) and complement component C5a).

**Figure 2 ijms-23-15823-f002:**
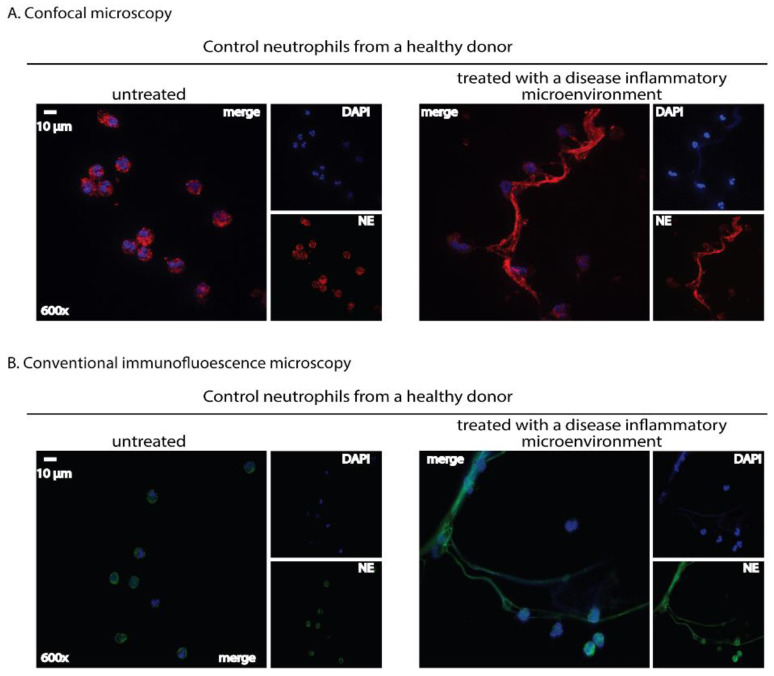
Immunofluorescence staining: A qualitative tool for NET detection in human-isolated neutrophils using either a confocal or a conventional fluorescence microscope. Neutrophils isolated from healthy donors’ whole blood using a density gradient separation method were stimulated with a serum derived from a treatment-naïve patient with active ulcerative colitis (180 min incubation time). NETs are visualized as fibers of extracellular DNA and neutrophil granule proteins assessed by double-staining with a chromatin-staining dye and a neutrophil marker. (**A**) Neutrophils were stained with a rabbit anti-human neutrophil elastase ((NE), Abcam, Cambridge, UK) antibody detected by a goat anti-rabbit AlexaFluor 594 antibody (Invitrogen, Waltham, MA, USA). Visualization was performed using confocal microscopy (Revolution spinning disk confocal system; Andor, Belfast, UK). (**B**) Neutrophils were stained with a mouse anti-human neutrophil elastase ((NE), Abcam) antibody detected by a goat anti-mouse AlexaFluor 488 antibody (Invitrogen). Visualization was performed using a fluorescence microscope (OLYMPUS BX51, Shinjuku, Japan). In (**A**,**B**), cell nuclei were counterstained with 4′,6-diamidino-2-phenylindole ((DAPI), Sigma-Aldrich, Burlington, MA, USA). Untreated neutrophils served as control. Data from the Molecular Hematology Laboratory archive are shown.

**Figure 3 ijms-23-15823-f003:**
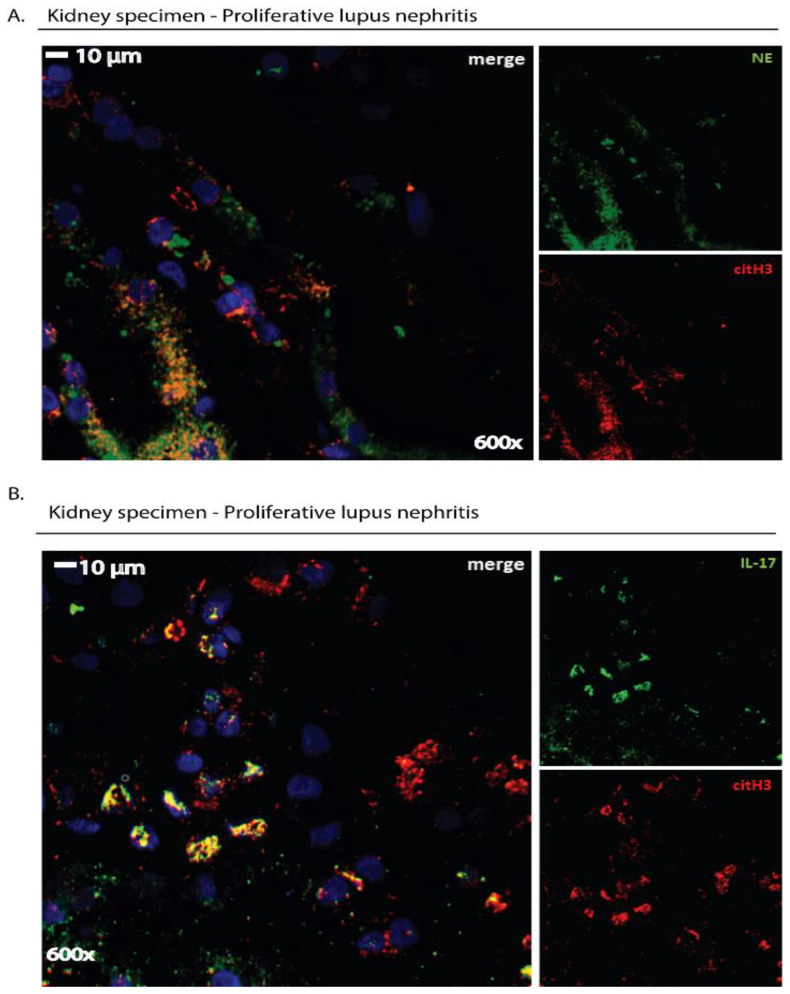
Immunofluorescence staining: a qualitative tool to detect NET remnants or disease-related proteins located on NETs in human tissue specimens. Cross-sections (4 μm thickness) of kidney biopsy tissue derived from a patient with proliferative lupus nephritis (LN) were deparaffinized and stained with the appropriate antibodies. (**A**) NETs were visualized in kidney specimens as extracellular structures by staining with both a mouse anti-human neutrophil elastase ((NE), Abcam) antibody and a rabbit anti-human histone 3 (Anti-Histone H3 (citrulline R2 + R8 + R17), Abcam) antibody. (**B**) The presence of a disease-related protein on extracellular structures, such as IL-17 in LN, was examined using both a mouse anti-human IL-17A (R&D Systems) antibody and a rabbit anti-human histone H3 (Anti-Histone H3 (citrulline R2 + R8 + R17), Abcam) antibody. In (**A**,**B**), a goat anti-mouse AlexaFluor 488 antibody (Invitrogen) and a goat anti-rabbit AlexaFluor 594 antibody (Invitrogen) were used to detect primary antibodies. Sections were counterstained with DAPI and visualized in a confocal microscope (Revolution spinning disk confocal system; Andor, Belfast, UK). Data from the Molecular Hematology Laboratory archive are shown.

**Figure 4 ijms-23-15823-f004:**
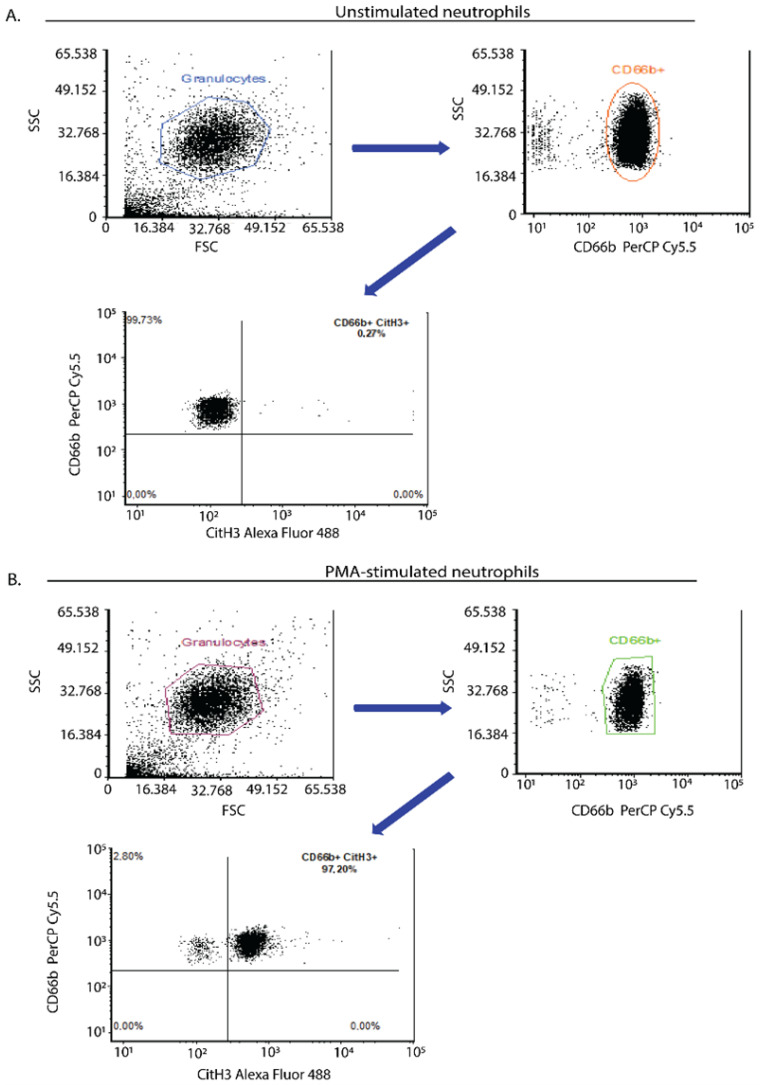
Flow cytometry assay: an indicative quantitative method of NET determination in human isolated neutrophils. Flow cytometric analysis of human neutrophils derived from a healthy donor. Neutrophils were isolated from whole blood using a density gradient separation method. (**A**) Unstimulated neutrophils (negative control) or (**B**) neutrophils stimulated with a chemical inducer of NETs (phorbol 12-myristate 13-acetate (PMA), Sigma-Aldrich; 20 nM final concentration, 45 min incubation time) were stained with a neutrophil activation marker (mouse anti-human CD66b—conjugated with PerCP/Cyanine5.5, Biolegend, San Diego, CA, USA). CD66b positive cells were examined for the presence of citrullinated histone 3 upon staining with a rabbit anti-human histone 3 antibody (Recombinant anti-histone H3 (citrulline R8), Abcam) detected with a goat anti-rabbit AlexaFluor488 antibody (Invitrogen). It was tested that PMA did not induce apoptosis/necrosis in neutrophils under these experimental conditions. Data from the Molecular Hematology Laboratory archive are shown.

**Table 1 ijms-23-15823-t001:** The benefits and the limitations of each laboratory method used in the context of NET evaluation.

Techniques	Benefits	Limitations	References
Enzyme-linked immunosorbent assay (ELISA)	Simplicity, specificity, cost-effectiveness, objective and quantitative	Results standardization and differences between in vivo and in vitro NETs	Kasprzycka et al., 2019 [41]; Thålin et al., 2017 [45]; Masuda et al., 2017 [49]
Immunofluorescence microscopy (IFM)	Distinguishes cell death mechanisms, objective quantification of NETs with software-based methods, and detects NETs in vitro, ex vivo and in situ	A time-consuming procedure, laborious, limited reproducibility, not available for rapid screening of many cells or samples, observer-dependent, and extensive precaution with sample preparation	Brinkmann et al., 2004 [11]; Von Köckritz-Blickwede et al., 2010 [50]; Brinkmann et al., 2012 [51]; Coelho et al., 2015 [52]; De Buhr and von Köckritz-Blickwede, 2016 [15]; Lv et al., 2020 [53]
Electron microscopy	Distinguishes cell death mechanisms and detects NETs in vitro and in situ	Biased based on the field of view and no reliable distinguishing between NETs and fibrin by SEM	Fuchs et al., 2007 [24]; De Buhr and von Köckritz-Blickwede, 2016 [15]; Krautgartner et al., 2008 [54]; Lv et al., 2020 [53]
Live imaging	Live in vivo imaging using animal models, close examination of NET structures, and distinguishes NETs from other cell deaths	Staining of many components on NETs, complex equipment and expensive	Cho et al., 2011 [55]; Kolaczkowska et al., 2015 [56]; Gupta et al., 2018 [57];
Flow cytometry	Qualitative, quantitative and objective; detects pre-NETotic cells, both in vivo and in vitro, rapid and sort between cell populations and has the possibility of direct applications in blood samples	Extensive precaution with sample preparation; detects only pre-NETotic cells and is incapable of detecting released NETs/remnants	Masuda et al., 2017 [49]; Kasprzycka et al., 2019 [41]; Gavillet et al., 2015 [58]; Zharkova et al., 2019 [59]
Multispectral image flow cytometer (MIFC)	Rapid, semiautomated and studies subcellular morphology and distinguishes cell death mechanisms	Sophisticated equipment and detects fewer NETotic cells than existing	Zhao et al., 2015 [60]
Western blot	Semi-quantitative, specificity and sensitivity	Multi-step protocol and optimizes the concentration of the primary antibody	Liu et al., 2016 [61]; Lv et al., 2020 [53]

## Data Availability

Not applicable.

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
