# Peer review of "Methods for the Assessment of NET Formation: From Neutrophil Biology to Translational Research"

_ijms, 2022, doi:10.3390/ijms232415823_

Round 1

Reviewer 1 Report

This article review compares the most commonly used techniques for the assessment of NET formation and release. The role of NET in a wide range of inflammatory and thrombotic disorders highlight their importance as possible therapeutic target or/and a diagnostic tool. The review provide some important insights into the qualitative and quantitative molecular analysis of NETs in translational medicine today.

Lines 38-39...and they have longer survival time than the one initially suggested [7] the authors can specify un example

Line 45 .... their capacity to release NETs [11]. It will be better to specify and say:  .... their capacity to release NETs as a new way to Kill Bacteria [11].

Lines 51-52 it should be better to distinguish physiological stimuli, as cytokines, pathogens, LPS etc, and chemical compounds that “in vitro” can induce NET formation.

Fig 1 it should be better clarify the role of Ca2+ which is also necessary es [18] for mt ROS generation or for NADPH activation.

Lines 56-57 ... DNA, citrullinated histones (CitH3), and other granular proteins are released.... (Figure 2). Other can be removed, in addition Figure 2 doesn’t show CitH3..a figure with CitH3 should be added.

The legend to Figure 2 should be made more fluid, for example: Neutrophils.... were stimulated for 180 min with serum derived....and so on throughout the text.

Line 58: Since NET formation was first reported in 2004 as an eliminating mechanism for invading microbes [20], is the reference correct? Or is [11]?

Lines 60-61 Aggregated NETs contribute to degrading cytokines and chemokines, resulting inflammation resolution [21]. The authors can specify in gout acute inflammatory reaction. It is good for the reader to know briefly what you are talking about without necessarily looking for the reference.

Lines 64-65 the sentence .. Neutrophils express on NETs a broad repertoire of disease-related proteins critically implicated in the process of several human disorders..; the sentence is repetitive.

Line 66 and beyond...formation of autoantibodies [23] and to act as a functional scaffold for thrombosis [24]. as specified before (lines 60-61) the authors can briefly detail giving some examples.

Line 86, It is not agreed upon which is the most targeted biomarker for detecting NET... Should be better It is not agreed upon which is the most targeted best biomarker for detecting NET...

Lines 87-88 , ...as their source can be from an irrelevant procedure, such as apoptosis or necrosis....

Is not clear, do you mean .....as they can derive from necrotic or apoptotic cells?

Line 94 ... [44] are considered targeted NET biomarkers.  Can be added: methods that detect the presence of two markers are certainly better however...

Line 99 .. as it provides simplicity [41]. Should be instead.  easy to use

Line 109 Immunofluorescence Microscopy, the order of methods is not the same as the table, so it is recommended to modify the table or the text

Line 113 .. to visualize the location of the DNA, is better .. to visualize DNA

Lines 118-119 ... in vivo in blood samples and fixed tissue specimens should be better: ex vivo in blood samples and fixed tissue specimens

Lines 129-130 the terms decorated : NETs are also decorated with a variety of proteins closely related with the pathophysiology of each disorder (“disease-related” proteins) [15] in my opinion is normally used for granules enzymes that characterize NETs, since their discovery and not for particular disease -associated proteins, the authors must better refer to the use of this term. Alternatively they can say NETs are also enriched with disease -associated proteins.

Line 134 both antibodies should be substituted by two antibodies

Lines 137-138 the method from Brinkmann et al., is not clear and should be better explained

Line 139 fluorescent microscope is not correct: fluorescence microscope

Lines 180-181 interesting role between NETs and platelets, probably you mean interesting connection between NETs and platelets

Line 184 explain the term CPAF, for example: the chlamydial protease CPAF

Line 209 What .. due to overlapping antibodies.. does it mean?

Line 232 ...dye, antibodies… it is better to say …dye, conjugated antibodies..

Line 255.. in human it is better to say …of human

Line 257 what is the PMA concentration?

Line 266.. information about the nuclear.. the meaning of this sentence is unclear

Line 269 .. reproducibility on the results.. … it is better to say …… reproducibility of results …

Line 272 netting neutrophils mean in early phase of NET formation?

Line 290 .. These NETs constitute the source of organ damage. The sentence is to strong and should be better to say .. These NETs could represent the source of organ damage

Lines from 303 DNA dyes

A rapid and easy method to detect NETs in a plate assay employs impermeable DNA dyes, such as Sytox Green, even after... An impermeable DNA dye (e.g Sytox Green) ...and .. A permeable DNA dye (e.g pico-Orange). But later it says: (line 310) Sytox Orange, an impermeable dye and (line 312) both an impermeable (Sytox Orange) and a permeable dye (Sytox Green), in cathalogs is indicated as impermeant to live cells.

Line 316 what is the meaning of.. swarming the fungi..?

In conclusion the review provides reliable and efficient  tools for ΝΕΤ identification and describes benefits and limitations for each method reaching the conclusion that there is not a gold standard method for assessment of NET formation.

The topic is actual and interesting, many methods are described in an exhaustive manner. However some sentences are unclear, sometimes it would be necessary to specify the statements reported.

This may also be partly due to incorrect English which often gives rise to ambiguous statements, so

the language must be revised throughout the text

Author Response

REVIEWER 1: This article review compares the most commonly used techniques for the assessment of NET formation and release. The role of NET in a wide range of inflammatory and thrombotic disorders highlight their importance as possible therapeutic target or/and a diagnostic tool. The review provide some important insights into the qualitative and quantitative molecular analysis of NETs in translational medicine today.

Point-by-point responses 

Lines 38-39...and they have longer survival time than the one initially suggested [7] the authors can specify un example

An example has been added.

Line 45 .... their capacity to release NETs [11]. It will be better to specify and say:  .... their capacity to release NETs as a new way to Kill Bacteria [11].

This specification has been added.

Lines 51-52 it should be better to distinguish physiological stimuli, as cytokines, pathogens, LPS etc, and chemical compounds that “in vitro” can induce NET formation.

This point has been clarified.

Fig 1 it should be better clarify the role of Ca2+ which is also necessary [18] for mt ROS generation or for NADPH activation.

We have modified Figure 1, since Ca2+ seems to be necessary for each type of NET formation. In addition, appropriate phrases have been incorporated in the revised manuscript (lines 50-51 and 57-58).

Lines 56-57 ... DNA, citrullinated histones (CitH3), and other granular proteins are released.... (Figure 2). Other can be removed.

The word has been removed.

In addition Figure 2 doesn’t show CitH3..a figure with CitH3 should be added.

Unfortunately, we have no unpublished data with high quality to provide. Hence, we decided to present this panel in which neutrophil elastase (NE) was used as a neutrophilic marker and DAPI was used to stain the extracellular DNA. This double-staining verifies the formation of NETs.

The legend to Figure 2 should be made more fluid, for example: Neutrophils.... were stimulated for 180 min with serum derived....and so on throughout the text.

Appropriate changes have been incorporated in the legend of Figure 2. Α corresponding change has been also made in the legend of Figure 3.

Line 58: Since NET formation was first reported in 2004 as an eliminating mechanism for invading microbes [20], is the reference correct? Or is [11]?

We apologize for this mistake. The reference has been corrected.

Lines 60-61 Aggregated NETs contribute to degrading cytokines and chemokines, resulting inflammation resolution [21]. The authors can specify in gout acute inflammatory reaction. It is good for the reader to know briefly what you are talking about without necessarily looking for the reference.

We apologize for this omission. The necessary addition has been made.

Lines 64-65 the sentence. Neutrophils express on NETs a broad repertoire of disease-related proteins critically implicated in the process of several human disorders; the sentence is repetitive.

The sentence has been removed.

Line 66 and beyond...formation of autoantibodies [23] and to act as a functional scaffold for thrombosis [24]. as specified before (lines 60-61) the authors can briefly detail giving some examples.

Brief examples have been added.

Line 86, It is not agreed upon which is the most targeted biomarker for detecting NET... Should be better It is not agreed upon which is the most targeted best biomarker for detecting NET.

The sentence has been changed.

Lines 87-88, ...as their source can be from an irrelevant procedure, such as apoptosis or necrosis....Is not clear, do you mean .....as they can derive from necrotic or apoptotic cells?

The sentence has been reworded to be clear.

Line 94 ... [44] are considered targeted NET biomarkers.  Can be added: methods that detect the presence of two markers are certainly better however...

The sentence has been changed.

Line 99 .. as it provides simplicity [41]. Should be instead.  easy to use

The sentence has been changed.

Line 109 Immunofluorescence Microscopy, the order of methods is not the same as the table, so it is recommended to modify the table or the text

The order of the methods in Table 1 have been modified.

Line 113 .. to visualize the location of the DNA, is better .. to visualize DNA

The sentence has been changed.

Lines 118-119 ... in vivo in blood samples and fixed tissue specimens should be better: ex vivo in blood samples and fixed tissue specimens

The correction has been done.

Lines 129-130 the terms decorated: NETs are also decorated with a variety of proteins closely related with the pathophysiology of each disorder (“disease-related” proteins) [15] in my opinion is normally used for granules enzymes that characterize NETs, since their discovery and not for particular disease -associated proteins, the authors must better refer to the use of this term. Alternatively they can say NETs are also enriched with disease -associated proteins.

It has been rephrased.

Line 134 both antibodies should be substituted by two antibodies

It has been substituted.

Lines 137-138 the method from Brinkmann et al., is not clear and should be better explained

Further details for the method have been introduced.

Line 139 fluorescent microscope is not correct: fluorescence microscope

It has been corrected.

Lines 180-181 interesting role between NETs and platelets, probably you mean interesting connection between NETs and platelets

The phrase has been changed.

Line 184 explain the term CPAF, for example: the chlamydial protease CPAF

The appropriate explanation has been introduced.

Line 209 What .. due to overlapping antibodies.. does it mean?

We apologize since the wording of this sentence was incomplete. We referred in in situ experiments. Hence, in these cases, there may be overlap in dyes due to the presence of various cell populations. The sentence has been corrected.

Line 232 ...dye, antibodies… it is better to say …dye, conjugated antibodies..

The sentence has been changed.

Line 255.. in human it is better to say …of human

The sentence has been changed.

Line 257 what is the PMA concentration?

The concentration of PMA has been introduced.

Line 266.. information about the nuclear.. the meaning of this sentence is unclear

It has been rephrased to better explain the advantages of this method.

Line 269 .. reproducibility on the results.. … it is better to say …… reproducibility of results …

The sentence has been changed.

Line 272 netting neutrophils mean in early phase of NET formation?

Yes, we referred in the early phase of NET release. The point has been clarified.

Line 290 .. These NETs constitute the source of organ damage. The sentence is to strong and should be better to say .. These NETs could represent the source of organ damage

We apologize for this intense positioning. The sentence has been corrected.

Lines from 303: DNA dyes.. A rapid and easy method to detect NETs in a plate assay employs impermeable DNA dyes, such as Sytox Green, even after... An impermeable DNA dye (e.g Sytox Green) ...and .. A permeable DNA dye (e.g pico-Orange).

But later it says: (line 310) Sytox Orange, an impermeable dye and (line 312) both an impermeable (Sytox Orange) and a permeable dye (Sytox Green), in cathalogs is indicated as impermeant to live cells.

We apologize for this typo error. It has been changed as “permeable dye (Cyto Green)”.

Line 316 what is the meaning of.. swarming the fungi..?

A brief explanation has been added.

Reviewer 2 Report

The review on the detection and analysis of NETS is thorough and sufficiently detailed and comprehensive for publication. The writing is also clear, and the manuscript is well organized. The advantages, relative costs, and disadvantages are discussed and compared. I recommend that the manuscript be published as it is. 

Author Response

We wish to thank the Reviewer 2 for his/her contribution.

Round 2

Reviewer 1 Report

The review was implemented with the indicated corrections, however English correction throughout the text was not done.

In addition the requested correction concerning the misuse of sytox green as a permeable nuclear dye instead of a non-permeable was contradictory: in fact now in the text (Line 328) Sitox green was replaced with Sytox Green, while in the section author response to report the authors write: "We apologize for this typo error. It has been changed as “permeable dye (Cyto Green)”.

Author Response

REVIEWER 1

Point-by-point responses 

The review was implemented with the indicated corrections, however English correction throughout the text was not done.

  • We apologize for this omission. English revision has been done throughout the text to improve any grammatical and syntactic errors.

In addition the requested correction concerning the misuse of sytox green as a permeable nuclear dye instead of a non-permeable was contradictory: in fact now in the text (Line 328) Sitox green was replaced with Sytox Green, while in the section author response to report the authors write: "We apologize for this typo error. It has been changed as “permeable dye (Cyto Green)”.

  • In the Review article, we indicatively discuss about two specific dyes used to distinguish dead from live cells i.e, SYTOX Orange and Syto Green. SYTOX Orange is a cell-impermeant nucleic acid stain. On the other hand, Syto Green is a cell-permeant nucleic acid stain. Accordingly, we refer in the manuscript (page 13, lines 333-335) that: “Of course, they still used microscopy techniques with both an impermeable (Sytox Orange) and a permeable dye (Syto Green) to verify the induction of NETosis which was further verified with a marker for NE [89].”